# From Science to Policy and Practice: A Critical Assessment of Knowledge Management before, during, and after Environmental Public Health Disasters

**DOI:** 10.3390/ijerph16040587

**Published:** 2019-02-18

**Authors:** Mélissa Généreux, Marc Lafontaine, Angela Eykelbosh

**Affiliations:** 1Eastern Townships Integrated University Centre in Health and Social Services—Sherbrooke Hospital University Centre, Sherbrooke, QC J1G 1B1, Canada; 2Department of Community Health Sciences, Faculty of Medicine and Health Sciences, Université de Sherbrooke, Sherbrooke, QC J1H 5N4, Canada; 3Chemical Emergency Preparedness and Response Unit, Environmental Health Science and Research Bureau, Healthy Environments and Consumer Safety Branch, Health Canada, Ottawa, ON K1A 0K9, Canada; marc.lafontaine@canada.ca; 4National Collaborating Centre for Environmental Health, Vancouver, BC V5Z 4C2, Canada; angela.eykelbosh@bccdc.ca

**Keywords:** knowledge transfer, knowledge management, environmental public health, disaster risk management

## Abstract

Canada regularly faces environmental public health (EPH) disasters. Given the importance of evidence-based, risk-informed decision-making, we aimed to critically assess the integration of EPH expertise and research into each phase of disaster management. In-depth interviews were conducted with 23 leaders in disaster management from Canada, the United States, the United Kingdom, and Australia, and were complemented by other qualitative methods. Three topics were examined: governance, knowledge creation/translation, and related barriers/needs. Data were analyzed through a four-step content analysis. Six critical success factors emerged from the analysis: blending the best of traditional and modern approaches; fostering community engagement; cultivating relationships; investing in preparedness and recovery; putting knowledge into practice; and ensuring sufficient human and financial resources. Several promising knowledge-to-action strategies were also identified, including mentorship programs, communities of practice, advisory groups, systematized learning, and comprehensive repositories of tools and resources. There is no single roadmap to incorporate EPH expertise and research into disaster management. Our findings suggest that preparation for and management of EPH disaster risks requires effective long-term collaboration between science, policy, and EPH practitioners at all levels in order to facilitate coordinated and timely deployment of multi-sectoral/jurisdictional resources when and where they are most needed.

## 1. Introduction

Canada, like many countries, increasingly faces emergencies and disasters that have public health impacts [1], including large-scale chemical incidents (e.g., the 2013 Lac-Mégantic train derailment) and natural disasters (e.g., the 2016 Fort McMurray wildfires). Such environmental public health (EPH) disasters may cause extensive environmental, human, and material losses, and may sometimes affect entire communities and/or necessitate evacuation and relocation. In addition to acute health risks, a large body of literature indicates that the population burden of psychopathology in the aftermath of EPH disasters is substantial and potentially of long duration [2,3,4,5]. EPH disasters may differ from other public health emergencies (e.g., Ebola outbreak, pandemic influenza), as most require both a short-term response (within hours), as well as a longer-term response (including monitoring, remediation and/or restoration efforts) that may stretch over years. And generate a need for multidisciplinary scientific expertise (chemistry, epidemiology, human health risk assessment, mental health, etc.). The increasing frequency and severity of EPH events is thought to be driven by the interactions of complex phenomena such as population and economic growth, land-use, resource scarcity, urbanization, and climate change, all of which are expected to continue into the foreseeable future.

The governance required to prepare for, respond to, and recover from a wide range of EPH disasters—natural or human-induced—is arguably the most complex and critical function of disaster management. Such EPH governance should serve to facilitate and strengthen capacity for risk assessment, surveillance, risk management, public communication, monitoring and evaluation, and mitigation and recovery activities. The components of EPH governance should, therefore, include not just policies, programs, and coordination structures, but should also address gathering and interpretation of relevant and up-to-date information with which to guide action [6].

### 1.1. Background

As the successor to the Hyogo Framework for Action 2005–2015: Building the Resilience of Nations and Communities to Disasters (HFA), the Sendai Framework for Disaster Risk Reduction 2015–2030 (Sendai Framework), adopted by 187 Member States at the Third United National World Conference on Disaster Risk Reduction, has shifted its emphasis from disaster management to disaster risk management [7]. With 35 explicit references to health, this people-centered framework encourages both risk reduction and resilience strengthening through an all-hazard, all-of-state and all-of-society approach [8]. Science should routinely be used to support disaster risk reduction [9] and, therefore, holds a key place in the Sendai Framework. Knowledge flowing from and to different stakeholders ensures that policy and practice are evidence-based and risk-informed (Figure 1; [10]).

### 1.2. Knowledge-to-Action (KTA) Process

Evidence-informed decision-making, in emergency management and in all other areas of public health, requires effective knowledge translation to turn research knowledge into action. The Canadian Institutes of Health Research (CIHR) have proposed a cyclical knowledge-to-action (KTA) process [11,12]. This process has been divided into various phases, from the identification of the problem to monitoring and sustaining the use of the available pertinent knowledge. The funnel, located at the heart of this cycle, represents knowledge creation. As it moves through the funnel, knowledge is refined and becomes more easily applicable to end-users. However, once created, knowledge needs to be translated into action. This cycle provides a framework for strategies in knowledge creation and translation.

Various types of knowledge may be utilized within KTA strategies in disaster settings. These include knowledge generated through science, as well as local (community-based) or indigenous knowledge [13]. Too often disaster management practitioners tend to focus on the former, ignoring the latter which is, nevertheless, of great value for disaster management, given local knowledge may provide contextual information not found in science-based information sources. For its part, scientific knowledge refers to primary literature (e.g., first-generation knowledge), knowledge synthesized from literature review (e.g., second-generation knowledge), and also to more user-friendly tools and resources, including decision aids, training modules, practice guidelines, lessons learned, and protocols (e.g., third-generation knowledge) [14]. Knowledge can also be classified as either tacit (i.e., understood or implied knowledge that exists without being stated) or explicit (i.e., formal or codified knowledge that is stated in detail) [15]. Such concepts build notably on the “knowledge creation spiral” theory within organizations, introduced by Nonaka and Takeuchi in 1995, which emphasizes on the importance of involving both the top and front-line employees in knowledge creation process [16]. By integrating top-down and bottom-up approaches, this system enables the creation, accumulation, and translation of tacit and explicit knowledge.

Effective KTA processes must also account for how knowledge is shared and transformed through sharing. Four processes transforming knowledge from one form to another have been identified, namely:Socialization;Externalization;Internalization; andCombination.

Socialization consists of sharing of individual tacit knowledge through collaborative methods, like meetings (tacit to tacit). Externalization refers to codifying the tacit knowledge into tools and resources (tacit to explicit). Internalization corresponds to learning by doing, through simulations or exercises (explicit to tacit). Finally, combination can be defined as the extraction and the combination of explicit knowledge, in order to organize it into various forms, such as repositories (explicit to explicit) [17].

### 1.3. Toward a National Framework

Canada has gained significant expertise and knowledge from recent EPH disasters, including the 2013 Lac-Mégantic train derailment, the 2016 Fort McMurray wildfires, and the 2016 Seaforth Channel diesel spill [18,19,20,21]. However, despite this growing expertise, two fundamental challenges remain. First, how can we improve knowledge generation (through research) before, during, and after disasters? Although disasters mobilize public health practitioners in a matter of hours, organizing scientific research efforts, and/or acquiring access to the knowledge generated may take weeks or months. Second, how can we improve dissemination and use of new and existing knowledge? These are especially complex challenges in Canada, in which knowledge-generating entities must cooperate across three levels of government, two official languages, and vast geographic distances. Rather than reinventing the wheel time and time again, how can we ensure that public health actors at all levels know what has been done elsewhere, how can we understand the uncertainties, and how can we integrate it in the face of a disaster?

Given the short time-frames and potentially high human costs that characterize EPH disasters, a national framework is urgently needed to facilitate and integrate knowledge creation and associated research into emergency response and recovery in Canada. Such a framework would allow public health professionals to systematize what has already been done, build on existing assets—such as expertise, evidence, guidelines, lessons learned, resources, training and protocols—and clarify what remains to be achieved through future collaborations.

Our aim was to determine how to better integrate EPH knowledge and assets in disaster settings in Canada [22], using the Sendai Framework as a template [23]. To that end, we conducted a critical assessment of knowledge management (i.e., providing the right information, in the right place, at the right time) before, during, and after EPH disasters in Canada. The following specific objectives were pursued to achieve this goal:Describe various existing models of governance for disaster management, with a focus on the science-policy-practice interface;Identify main resources available and challenges for knowledge management; andFormulate recommendations toward the establishment of a national framework.

## 2. Materials and Methods

### 2.1. Design

This project draws heavily on the US National Institutes of Health (NIH) Disaster Research Response (DR2) Program. The DR2 Program is a national framework for research on the medical and public health aspects of disasters and public health emergencies [24,25] which offers data collection tools and resources, training and exercise materials, and research protocols, and facilitates networking between researchers and practitioners responding to environmental emergencies. This project was led in close collaboration with a national steering committee created in 2016, to oversee the development of a Canadian DR2 (CanDR2); the committee is co-chaired by Health Canada and the National Collaborating Centre for Environmental Health (NCCEH) and is composed of representatives from many Canadian and international agencies.

### 2.2. Sample Selection

Key informants (KIs) involved in preparedness for, response to, and recovery from EPH disasters, both at the local and national (i.e., provincial or federal) levels in Canada and other countries were invited to participate in this initiative.

The KIs identified were experienced emergency planners from health or non-health sectors, medical officers of health, knowledge transfer experts, or academics, who possessed expertise in the management of natural (e.g., floods, winter storms, heat waves, hurricanes, wildfires) or technological (e.g., chemical spills, train derailments) disasters.

At least one representative from each of the provinces of British Columbia, Alberta, Ontario, Québec, and the Atlantic Canada region was sought. These provinces or regions were selected based on their geographic location (dispersed across Canada), their experiences in facing EPH disasters, and the varying size and capacity of their public health workforce. In addition, KIs from the federal level in Canada were invited to participate in this initiative, as well as KIs from governmental or non-governmental organizations in other key English-speaking countries, including the United States, the United Kingdom, and Australia. While we emphasized the importance of diversity across KIs (at all levels, from governmental and non-governmental organizations, from health and non-health sectors), this diversity did not prevent our study from reaching data saturation (i.e., we obtained a sample large enough such that no new data were generated from additional participants).

An initial list of KIs was generated by the CanDR2 Steering Committee and invited to participate through an introductory email briefly describing the project. Nearly all invitees accepted, with a final sample of 23 KIs.

### 2.3. Data Collection Instrument and Methods

An interview guide containing mostly open-ended questions deemed to be relevant to our objectives was developed and endorsed by the members of the CanDR2 Steering Committee. Four overarching dimensions of the EPH response to disasters were explored, namely (1) governance, (2) knowledge creation, (3) knowledge translation, and (4) barriers and needs related to these processes [26]. The interview guide drew heavily on the World Health Organization (WHO)’s toolkit for assessing health system capacity for crisis management, more specifically on the governance and leadership function. This standardized toolkit is organized according to the six functions of the WHO health system framework. For the first function, that is effective leadership and governance, fourteen essential attributes are described, including programs on preparedness, and research and evidence base [6].

The interview process comprised two parts. In the first part of the interview, seven questions were used to broadly characterize the governance model/strategic framework in each of the jurisdictions under study, in order to obtain an overview of disaster management structures and coordination mechanisms. The second part of the interview specifically addressed knowledge creation (three questions) and knowledge translation processes (five questions). Existing barriers and factors conducive to strengthening of science-policy-practice interface and integration of knowledge into action were assessed throughout the interviews. A specific question on obstacles and strengths was also added at the end of the guide. Each interview lasted approximately 60 min. Interviews were conducted by the project leader in English (*n* = 16) or in French (*n* = 7), either on the telephone (*n* = 19) or face-to-face (*n* = 4), from April to July 2017.

### 2.4. Other Data Sources

In-depth interviews were complemented by other qualitative methods. The project leader was involved as the public health director or a medical adviser during three large-scale EPH disasters in Canada, namely the 2013 Lac-Mégantic train derailment, the 2016 Fort McMurray wildfires, and the 2017 Quebec flood. Furthermore, she attended 15 scientific fora (conferences, workshops, symposia, etc.) related to EPH disaster management from July 2016 to July 2017. Throughout these meetings and informal discussions with key international experts, additional explanations and good practices were documented. Field notes were taken after each scientific forum. Finally, a wide range of documents and websites suggested by KIs were consulted to deepen the understanding of laws, structure, policies, plans, procedures and programs identified during interviews.

### 2.5. Analysis

In order to carry out the analysis of the KIs’ discourse and other data gathered, we conducted a four-step content analysis: (1) double-reading of transcripts, (2) data coding, (3) data processing, and (4) interpretation of data.

In Step 1, an initial (appropriation) reading of interviews and field notes was used to identify the main ideas characterizing each interview or event. The second (reading served to confirm and clarify these ideas. In Step 2, raw qualitative data were classified into an analysis grid, with a coding structure based on four dimensions established a priori: governance, knowledge creation, knowledge translation, and barriers and needs [26]. In the data processing Step 3, the analysis grid was used to draw out themes and subthemes from the coded transcripts. Data that were conceptually related to one another were first identified within each dimension. This set of data became sub-themes corresponding to ideas or concepts in relation with our subject. Examination of these sub-themes across all four pre-determined dimensions then led to their grouping into cross-cutting themes. The final Step 4 was interpretation of the coded and processed data in the context of our initial objectives [27].

### 2.6. Integrated Knowledge Translation

In accordance with the principle of integrated knowledge translation, as promoted by CIHR [28], members of the CanDR2 Steering Committee were involved in each step of the project. Early in the process, they contributed in identifying the problem and clarifying the aim of the project, they were invited to comment on the protocol and the interview guide, and they identified a list of potential KIs. Once data were gathered and analyzed, preliminary and final findings were presented and discussed with members of the CanDR2 Steering Committee on two occasions. Presentation of findings and recommendations to various stakeholders (e.g., organizations involved in disaster management in Canada) is also planned in the upcoming months.

## 3. Results

### 3.1. Description of the Data

Overall, 16 interviews were conducted among KIs from Canada (*n* = 16) and other jurisdictions (*n* = 7) (see Table 1). Fifteen KIs came from the public health or health sector, whereas the remaining KIs came from the municipal sector (*n* = 4), academia (*n* = 2), or non-governmental organizations (NGO; *n* = 2). The sample had balanced representation at the national and local levels, and of both genders. Two KIs represented the indigenous communities’ perspective. Interestingly, many KIs had been involved in the management of EPH disasters, including the 2005 explosion and fire at the Buncefield oil storage depot (United Kingdom), the 2009 Black Saturday Bushfires in Victoria (Australia), the 2011 Alberta Slave Lake floods (Canada), the 2012 Neptune Technologies explosion (Canada), the 2013 Lac-Mégantic train derailment (Canada), the 2016 Fort McMurray wildfires (Canada), the 2016 Seaforth Channel diesel spill (Canada), Hurricane Matthew in 2016 (United States), the 2017 New Brunswick ice storm (Canada), the 2017 Quebec flood (Canada), the Flint water crisis (United States), as well as annual flooding in Ontario (Canada).

### 3.2. Emerging Themes: Critical Success Factors

Six cross-cutting themes, which are here identified as critical factors in successful disaster knowledge management, materialized from the data interpretation, with a range of sub-themes emerging in each category. These sub-themes, which represent the current situation and challenges, are further discussed below.

#### 3.2.1. Blending the Best of Traditional and Modern Approaches

The data revealed that, in Canada and other developed countries, disaster management is well structured at all levels and that, overall, public health authorities are involved in these institutional arrangements. Most structures rely on the incident command system (ICS) for coordinating disaster responses, consider all phases of the disaster management continuum (mitigation, preparedness, response, and recovery) and are supported by laws, policies, plans, and procedures. Routine surveillance and epidemiological investigations are fairly well integrated during the response phase, as are conferences, meetings, training activities, and exercises during the preparedness phase. In short, the above traditional approaches are adequately implemented.

More modern all-hazard approaches [29,30] as promoted in many key documents, including the Sendai Framework [7], are currently integrated into disaster management preparedness and response in some jurisdictions. However, for most KIs in Canada and elsewhere, the Sendai Framework is not known or not a priority. Some KIs nevertheless expressed their desire for a major paradigm shift: “Requirements are based on old emergency management approaches and principles, we need governance from a different perspective” (KI#6). Others emphasized the importance of better understanding the potential risks: “It is important to be involved in risk assessment before a disaster strikes, to better manage risks altogether, as proposed in the Sendai Framework” (KI#9). An observation made at the Fifth Regional Platform for Disaster Risk Reduction in the Americas [31], which was also noted by one KI, is the poor representation of local-level participants of some major countries at this meeting: “Those involved don’t share with lower levels, it’s the opposite than [sic] what’s proposed in the framework” (KI#18). This suggests that even if we are on the right track in implementing modern approaches, additional efforts are required to fully achieve this goal.

#### 3.2.2. Fostering Community Engagement

The importance of identifying and leveraging existing assets or resources at the community level, including local health agencies, and working with existing capacities were strongly valued among KIs. Furthermore, it was indicated that local knowledge should be given consideration in the same manner as scientific knowledge: “We need to hear more from the community, it’s really important” (KI#15). However, it was broadly acknowledged that communities typically remain poorly engaged in disaster management and that strategies to foster community engagement while maintaining the efficiency of disaster response and research are yet to be developed [32]. The interviewees indicated that ideally such strategies would provide a mixture of top-down and bottom-up approaches, and mobilize local knowledge and expertise before, during, and after a disaster.

Although leveraging local capacity was recognized as important, KIs also noted that building capacity in small municipalities, rural and Indigenous communities remains a challenge, and that support is required from higher-level governmental and academic institutions. For example, one KI (who has previously responded to a fuel spill) noted that more detailed technical guidance is needed to support local risk assessment: “What kind of environmental and biospecimen sample should be taken? What’s needed? How often? What parameters? What detection limits? What should be the benchmarks (or appropriate end-points)?” (KI#3). Developing this type of technical guidance is often beyond the capacity of those responding to the crisis at hand, as such work requires time, research, and multidisciplinary expertise.

#### 3.2.3. Cultivating Relationships

Many of the successes identified by KIs relied on individual leadership (or “champions”) and strong interpersonal connections. Beyond structures, plans, and procedures, formal or informal relationships and networks have been identified by the majority of KIs as the most promising avenue to strengthen knowledge management capacities in disaster settings. Indeed, many KIs emphasized the need for breaking down silos between sectors (e.g., first responders vs. public health), practitioners and academics, French-speaking and English-speaking provinces, local and national levels, and even within public health organizations [33]. For example, KI#16 noted that “First Nations communities can ask for support, but this depends a lot on relationships previously built with governmental agencies”. As noted by KI#9, “Strong links must be established right from the start between public health and first responders to establish a risk assessment and management strategy.” Because public health authorities are well connected with universities in many jurisdictions, it may be possible to build bridges with academics to bring additional expertise to affected communities.

Some KIs mentioned that an overlap exists between academic and public health expertise in the context of EPH disasters, in that both may offer expertise in epidemiology, toxicology, surveillance, risk, and exposure assessment, etc. These two essential partners, therefore, need to clarify their respective roles, in order to unite their efforts and act in a synergistic manner. Building such functional partnerships may unlock latent resiliency, especially if accomplished before a disaster strikes. As expressed by one KI, “We need to connect the dots. We need to better know each other before an event; it is too late to learn during the crisis” (KI#10). Indeed, the power of these “connected dots” was apparent during the response to the 2017 Quebec floods. Within 24 h, an international network of renowned organizations including the US Centers for Disease Control and Prevention (CDC), Public Heath England (PHE), and WHO were bringing concrete support to Quebec public health authorities. This pre-established network permitted rapid access to validated materials, including the Community Assessment for Public Health Emergency Response (CASPER) toolkit in the US [34] and a protocol and questionnaires from the National Study of Flooding and Health in the UK [35].

#### 3.2.4. Investing in Preparedness and Recovery

Although the four phases of disaster management are usually considered in disaster management structures, our data suggested local efforts are (by necessity) oriented toward the short-term. Our interviews suggested that Canadian EPH practitioners, particularly those at the local level, are struggling with one crisis activation after another; some organizations contacted during this study report remaining in near continual response mode in recent years. Consequently, little energy can be allocated to preparedness and recovery. Recovery is perhaps the most difficult task because of the accumulating burden on EPH professionals, including emotional load, fatigue, cumulative workload, and organizational factors, including less effective coordination structure and gradually weakening political commitment. According to one respondent, “we could probably do more between disasters, it is all about prioritization” (KI#5). As a result of this “disaster hangover”, long-term monitoring of physical and mental health issues is not routinely carried out, despite the fact that those issues probably have the most significant impact on populations [36].

Another critical area that is typically overlooked in the post-disaster landscape is the identification and sharing of problems and lessons learned. Once the disaster is over, teams are asked to return to their regular tasks as quickly as possible, such that “Practitioners don’t have enough time to think, to learn, to gain knowledge” (KI#11). As reported by many KIs, subsequent crises arise and similar problems are again encountered. For example, during the 2017 ice storm in New Brunswick (NB), one KI observed that public health was not as prepared as it could be, noting that “We always have to reinvent the wheel” (KI#12). This strongly expressed desire to improve learning from past events was the most consistent finding across KIs interviewed.

#### 3.2.5. Putting Knowledge into Practice

The interviewees reported the science-policy-practice interface before, during, and after EPH disasters is not as robust as it could be, at either the local or national level in Canada, or in the other countries considered. Several interviewees from local organizations, within and outside the health sector, mentioned working infrequently with researchers due to the demands of day-to-day operations. The lack of systematic mechanisms to incorporate research and expertise into disaster management was also noted. The first challenge identified was “making sure that we are asking the right questions to inform and to learn from a given event” (KI#2). While accessing local data (e.g., environmental monitoring, epidemiological investigations, victim registries, response information, and after action reports) is essential to generate knowledge, as suggested by some KIs, such sharing can be sensitive for partners involved in disaster response, including first responders, local authorities, and non-governmental organizations. Various cultural backgrounds, lack of clarity in respective roles and responsibilities, privacy issues, security clearance, fear of being judged, not having the mandate, and competition might explain this phenomenon. The search of those responsible and/or the causes after disasters is negatively perceived in many countries, which may become a growing obstacle to learning from previous experiences.

Once knowledge is generated, this newly developed evidence has to be transferred from experts and researchers to inform policies and practices. In the aftermath of the Fort McMurray wildfires, “an evaluation of the psychosocial response and recovery was conducted through focus groups and interviews to identify further needs and current gaps” (KI#4). These findings emphasize the importance of having trained, dedicated staff who are tasked with taking on these “additional” data collection and knowledge translation activities.

Another issue commonly reported is the need to adapt scientific knowledge to local context, to make it clear and concise. Ideally, more user-friendly tools and resources would be produced (i.e., third-generation knowledge): “There is a lot of knowledge, amazing amount of information, really good evidence. The challenge is having the time to make it digestible” (KI#15). The same issue has been raised regarding the Sendai Framework, which is much better known among academics than practitioners. Although “it is important, it is not clear how it will materialize” (KI#13). Finally, many local organizations are trying to develop a resource repository, but few have a comprehensive one yet. For example, although the current Canadian Disaster Database (https://www.publicsafety.gc.ca/cnt/rsrcs/cndn-dsstr-dtbs/index-en.aspx) facilitates learning by identifying events geographically and by types, Canada does not currently have a central repository to share resources (tools, reports, etc.) and connect seekers to the experts involved.

#### 3.2.6. Ensuring Sufficient Human and Financial Resources

Unsurprisingly, a recurring theme related to human and financial capacities. With few exceptions, disaster risk management and capacity-building activities are not prioritized, leading to insufficient funding and resources. According to many KIs, having trained, dedicated staff, either in local authorities or in specialized branches at the national level, could certainly help. Local emergency planners or coordinators have been found to play central roles in Canada and elsewhere. Although more focused on acute care than public health, Health Emergency Management BC (HEMBC) has been providing expertise, education, tools, and support since 2004 in BC, being “in charge of strategic planning, priority setting, networking, and performance measures” (KI#1). Another issue raised pertains to disaster research funding, which can be very complex. The first weeks following a disaster offer a window of opportunity that should be exploited, as political commitment and willingness to support research and long-term monitoring activities rapidly decrease thereafter. To secure funds, some KIs proposed that “disaster research should be seen as a national priority, with disaster-specific funding opportunities” (KI#11). Canada does not currently have an expedited research funding or ethical review process in place to address the immediate aftermath of disasters. For example, CIHR offered $2 million in health research funding after the Fort McMurray wildfires, but this grant competition did not start until October 2016 (i.e., a few months after the fires).

### 3.3. Promising Knowledge-to-Action Strategies

Throughout the multiple data sources examined, we discovered a wide range of effective solutions already adopted by many countries worldwide to promote the integration of current knowledge into emergency preparedness, response, and recovery. The objective of this project was not to undertake an exhaustive inventory of these solutions, but rather to draw attention to strategies that might be adaptable to the Canadian context. For each country surveyed, the most promising strategies have been categorized according to the four types of KTA processes identified by Rhem [17] (i.e., socialization, externalization, internalization, combination; see Table 2). Few, if any, innovative strategies were found with respect to the internalization process, which refers to learning by doing (e.g., drills or exercises). By contrast, a plethora of socialization strategies have been identified. These strategies, all more creative than the internalization strategies, have two common denominators. First, they draw on human capital. Second, whether they are deployed before, during, or after an EPH disaster, they all promote more effective interplay of science, policy, and practice.

Mentorship programs among employees and/or community members appear well established in the Canadian and Australian Red Cross organizations. For example, a mayor of a city previously devastated by a bushfire could in turn support another mayor currently facing a similar situation by sharing tools and lessons learned. Moreover, in the Netherlands and New South Wales (Australia), expert advisory groups can be activated to assist health authorities dealing with complex issues raised by environmental public health emergencies; depending on the situation, various types of expertise (e.g., environmental health, mental health, epidemiology, toxicology) can be mobilized within these jurisdictions to rapidly provide an overview of current scientific knowledge that might inform decision-making [37].

Various networks or communities of practice, both at national and international levels, have been put in place to better incorporate scientific and/or local knowledge into practice. An example of this is the WHO Thematic Platform for Health Emergency and Disaster Risk Management Research Group, a growing international network of policy-makers, practitioners, and researchers [38]. This initiative, recently underscored at the Global Platform in Cancun in May 2017 [39], is a good example of how the various stakeholders in disaster planning and management can all work together more effectively. An additional benefit of this global network whose members span all time zones is that the group can collectively support the response to a disaster occurring anywhere in the world at any time.

Externalization and combination processes were less frequently reported than socialization. However, two communities have released or are about to release reports on lessons learned after 2009 Victoria (Australia) bushfires [40] and the 2016 Seaforth Channel (Canada) diesel spill for public review and as learning resources.

Various approaches combining KTA processes have been developed. The USA has some of the most promising strategies, including “Lessons Learned Information Sharing”, a database set up by the Federal Emergency Management Agency (FEMA), and a comprehensive repository easily available on the National Library Medicine (NLM) website called Disaster Lit^®^. Through the latter, the NIH has put together 12,000 records of hand-picked documents related to public health disasters from the grey literature, including factsheets, guidelines, assessment tools, training material, reports, web pages, and web sites [41]. Another example is Evidence Aid, an initiative that aims to improve access to evidence on disaster-related health interventions, actions, and policies [42].

The city of Newcastle (UK) has demonstrated an exceptional degree of commitment to emergency/disaster preparedness and response. The city has hired three emergency planners who are fully dedicated to disaster management. Their work is undertaken in a multiagency space (including public health), with joint risk assessment and planning and a community risk register that drives the action plan. After every incident or exercise, a structured debrief is organized, as is often the case in many other jurisdictions; however, in Newcastle such information is also presented as a report to the city council and the “Overview and Scrutiny Committee” (led by the opposition), which together look at all of the council actions. According to KI#21, giving politicians ownership of the report makes the whole process open, transparent, and accountable. Local engagement and leadership has also led to the incorporation of other KTA strategies, including the recent organization of a conference entitled “Psychosocial Impacts of Emergencies” [43]. This event aimed to bring together people from multiple levels and different agencies to foster collaboration, promote community engagement, and raise awareness of the current gaps in disaster risk management.

## 4. Discussion

Authors translating knowledge to action by closing gaps between knowledge and practice is an iterative, dynamic, and complex process [12]. This project aimed to identify factors influencing this process, and explore solutions to promote the bridge between science, policy, and practice for disaster management. Our results corroborate and expand upon the published literature. Several studies have previously identified similar factors that may limit knowledge translation in disaster preparedness and response, including gaps in basic knowledge, such as the lack of long-term observational and interventional research [44,45,46], challenges in systematically collating and delivering lessons learned from events, and difficulties in creating and sustaining effective community engagement [47,48,49,50]. And as with other studies [15], we also found several factors facilitating the uptake of knowledge, including the importance that knowledge is tailored to local contexts and made actionable. Overall, our interview results support the promotion of interactions between researchers, experts, and users from all sectors to produce, disseminate, and make use of knowledge for the purpose of improving disaster preparedness and management [51,52,53].

More specifically, five recommendations emerged from the interviews with KIs and field observations:(1)Community of practice: A pan-Canadian community of practice involving emergency managers, public health practitioners, academics, local champions, Red Cross professionals, and any other stakeholders interested in EPH disasters, should be hosted within a trusted organization to support disaster preparedness. Within this community of practice, local initiatives could be shared and general consensus or understanding could be achieved regarding best practices in disaster response (e.g., risk assessment) and recovery (e.g., long-term monitoring). This would also be the ideal setting for the development of standardized tools for disaster health research as a basis for further action.(2)Roster of experts: Linked to the above community of practice, a roster of Canadian experts (e.g., researchers, toxicologists, epidemiologists, environmental health, occupational health, and mental health experts) should be created to support disaster response and recovery. We anticipate that this network could be called upon as needed to form scientific advisory groups to assist local authorities dealing with EPH disasters in both the short- and long-term. Such an initiative could also increase collaboration and sharing of expertise between researchers and EPH personnel in the field. Drawing on existing models (such as DR2), it would also lead to the identification of relevant research questions and the development of a research agenda that fits operational objectives [24,37,54].(3)Knowledge generation: A systematic mechanism to promote retention of learning from past events is required. All types of knowledge gained responding to previous disasters should be valued, whether this is first-, second-, or third-generation scientific knowledge, or local knowledge such as success stories, pilot initiatives, and lessons learned from the field [13]. As part of recovery operations, emergency managers and EPH practitioners should take the time to learn from their experiences and contribute to establishing a solid foundation upon which to can build national capacities. Debriefs should involve multiple sectors and seek the input from members of the community [55]. Standardized templates for after-action reports and a tracking system for correctable issues should be made available. Their use should be legislated after any exercise or real event in order to identify lessons, and most importantly to learn from them locally. The storage of completed templates in a central location (publicly available) would facilitate access to local knowledge and foster vertical and horizontal knowledge translation.(4)Knowledge transfer: There is an urgent need in Canada (and elsewhere) to gather and synthesize disaster-related knowledge, and to transfer it to other communities, ideally using same central space (i.e., a virtual repository) identified in item (3) above. This knowledge might take the form of research findings, research protocols, practice guidelines, data collection tools and resources, training and exercises materials, lessons learned, etc. Such a virtual repository could be developed de novo, but Canadian documents could also be identified and shared through the extensive repository of tools and resources available from the US (i.e., Disaster Lit^®^, an NLM resource guide). Moreover, Canadian representatives should be appointed to the NLM Disaster Information Specialist Program, which supports the provision of disaster-related health information resources to the disaster workforce through a network of information professionals and librarians.(5)Guidance on Sendai Framework: Guidance for a better integration of the Sendai Framework into health emergency management in the Canadian context should be developed. Such guidance would be particularly helpful for a growing number of stakeholders wishing a paradigm shift from disaster management to disaster risk management.

It is hoped our findings and recommendations will contribute to the identification and implementation of concrete solutions that foster the creation and the use of knowledge before, during, and after EPH disasters. Based on the challenges and successes identified during our interviews, we believe that there is no single roadmap to incorporate EPH expertise and research into disaster management. However, one thing is certain: these solutions should be developed not only during but also (or primarily) before and after disasters. The search for solutions should be based on the assumption that various types of knowledge translation processes are necessary [16]. Their complementary nature should be exploited in such a way as to develop a pan-Canadian framework that is adaptable to the needs of each province and territory, and other levels of government or sectors.

### Strengths and Limitations

This project is, to the best of our knowledge, the first to critically assess capacity for knowledge management in EPH disaster settings in Canada. Our diverse sample of KIs represented a wide range of Canadian perspectives: rural and urban, indigenous and non-indigenous, anglophone and francophone, and from the east coast to west coast. This sample was further complemented by respondents from other countries to capture perceptions, ideas, and experiences on a broader basis. The measuring instrument (i.e., the interview guide) was based upon a standardized toolkit developed by the WHO to assess capacities for crisis management [6]. Furthermore, examination of qualitative data followed a rigorous protocol that helped to increase the internal validity of the data collected for coding purposes. Finally, a steering committee composed of knowledge users from various backgrounds accompanied the entire process, from the identification of the problem to the validation of potential solutions.

There are inherent limitations to a qualitative approach based primarily on KI interviews. First, the sample size was limited to 23 KIs. Our interviews also gathered opinions and ideas that may have been influenced by many factors including past experiences, recall bias, social desirability, and hidden agendas. In order to minimize the effect of those factors, we used additional sources of information to complement the interview data. Our design has therefore facilitated triangulation of findings and improved both reliability and validity.

## 5. Conclusions

Generally speaking, critical success factors for public health action include good governance, development of strong and sound partnerships, dedicated capacity and resources, and use of evidence to inform actions. Good governance is perhaps the single most important factor influencing the effectiveness of emergency preparedness, response and recovery. Beyond structures and plans, it is necessary to cultivate relationships and share responsibility for ensuring the safety, health, and well-being of affected communities, while respecting the local culture, capacity, and autonomy. Preparation for and management of EPH disaster risks requires effective long-term collaboration between science, policy, and EPH practitioners at all levels in order to facilitate coordinated and timely deployment of multi-sectoral/jurisdictional resources when and where they are most needed.

## Figures and Tables

**Figure 1 ijerph-16-00587-f001:**
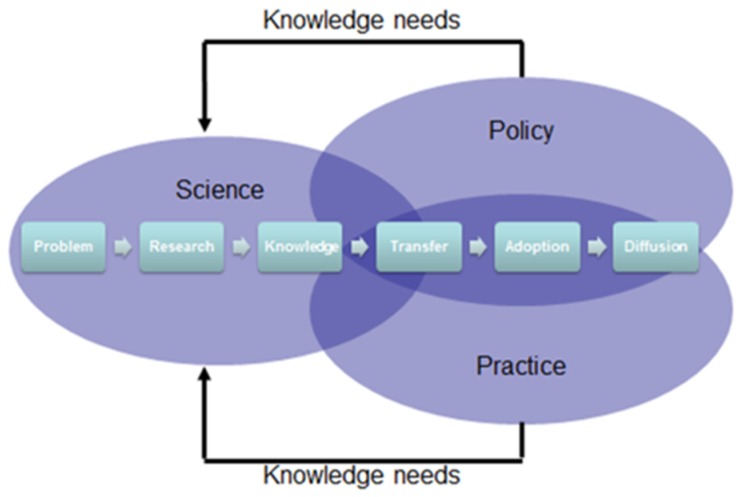
The science-policy-practice continuum (adapted from [10]).

**Table 1 ijerph-16-00587-t001:** Profiles of KIs interviewed in this study.

ID	Jurisdiction	Sector	Level	Gender
1	British Columbia	Health	National	F
2	British Columbia	Public Health	National	F
3	British Columbia	Public Health	Local	F
4	Alberta	Health	National	F
5	Alberta	Public Health	Local	M
6	Ontario	Health	National	M
7	Ontario	Public Health	Local	M
8	Québec	Municipal	Local	M
9	Québec	Municipal	Local	M
10	Québec	Municipal	Local	M
11	Québec	Academic	National	F
12	Atlantic	Public Health	Local	F
13	Canada	Public Health	National	F
14	Canada	Public Health	National	F
15	Canada	NGO	National	F
16	Canada	Public Health	National	M
17	United States	Public Health	National	F
18	United States	Academic	Local	M
19	United Kingdom	Public Health	National	F
20	United Kingdom	Public Health	National	F
21	United Kingdom	Municipal	Local	F
22	Australia	Public Health	Local	F
23	Australia	NGO	National	M

**Table 2 ijerph-16-00587-t002:** Examples of promising KTA strategies.

Socialization	Externalization	Combination
Canada		
Opportunities for professional growth from mentorships at Canadian Red Cross	Report on lessons learned by the community after the 2016 Seaforth channel spill	User-friendly Sharepoint^®^ with resources and tools shared on an ongoing basis in Alberta
Lessons learned from Slave Lake and Lac-Mégantic integrated into the mental health recovery plan in Fort McMurray	Book on the Lac-Mégantic tragedy sharing lessons learned by health and community networks	Environmental public health response and recovery toolkit in Alberta
Multisectoral debriefing after Neptune Technologie explosion that led to a better response in Lac-Mégantic	Mapping of responsibilities/accountabilities following recommendations at Canadian Red Cross	Emergency preparedness and response working group in NB to facilitate access to documents and resources
During 2017 Quebec floods, meeting with a city previously affected by a major flood to learn from past experiences		
During 2017 Quebec floods, visit of an expert on the ground to share his knowledge		
Provincial symposium organized by HEMBC		
United States		
Local emergency planning committees (federal mandate)	Rapid Needs Assessment facilitated by CASPER toolkit	Lessons learned database at FEMA
Phone call organized by CDC between 4 states affected after Hurricane Matthew	Central office for all after-action reports at CDC (problems and corrections)	Disaster Lit^®^: 12,000 records (grey literature) related to public health disasters at NLM
Midwest Consortium for Hazardous Waste Worker Training		NIH DR2 Program: Repository of surveys, questionnaires, protocols, guidance, forms
Environmental Justice Summit organized in Flint (Michigan)		
Disaster epidemiology community of practice		
Disaster information specialists at NLM		
United Kingdom		
Newcastle conference on psychosocial impacts of emergencies	Overview and Scrutiny Committee in Newcastle following after-action reports	Mapping of the Sendai Framework implementation: resources, projects, all sectors
Local resilience forums		
PHE Centre for Radiation, Chemical and Environmental Hazards		
UK Alliance for Disaster Research		
Australia		
Expert advisory panel/group activated by Chief Public Health Officer	Lessons from the community after 2009 Victoria bushfires	
Mentoring network at the Australian Red Cross		
The Netherlands		
Expert Group Health Research and Care after Disasters and Environmental Crises		
Global		
WHO Thematic Platform for Health Emergency and Disaster Risk Management Research Group	International Federation of Red Cross Psychosocial Center website: a lot of very useful resources	Global Public Health Intelligence Network (GPHIN): a web-based early-warning tool
UNISDR Scientific and Technical Advisory Group		Evidence Aid: reliable, up-to-date evidence on interventions in the context of emergencies
WHO collaborating center on chemical incidents		Weekly updates from the PHE Global Hazards Weekly Bulletin

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
