# Peer review of "From Science to Policy and Practice: A Critical Assessment of Knowledge Management before, during, and after Environmental Public Health Disasters"

_ijerph, 2019, doi:10.3390/ijerph16040587_

Reviewer 1 Report

 Given the importance of evidence-based, risk-informed decision-making, the authors aimed to critically assess the integration of EPH expertise and research into each phase of disaster management. In-depth interviews were conducted with 23 leaders in disaster management from Canada, the United States, the United Kingdom and Australia, and were complemented by other qualitative methods. Three topics were examined: governance, knowledge creation/translation, and related barriers/needs. Data were analyzed through a four-step content analysis. Six critical success factors emerged from the analysis: blending the best of traditional and modern approaches; fostering community engagement; cultivating relationships; investing in preparedness and recovery; putting knowledge into practice; and ensuring sufficient human and financial resources. Several promising knowledge to-action strategies were also identified, including mentorship programs, communities of practice,advisory groups, systematized learning, and comprehensive repositories of tools and resources.

There is no single roadmap to incorporate EPH expertise and research into disaster management. The findings of this paper suggest that preparation for and management of EPH disaster risks requires effective long-term collaboration between science, policy, and EPH practitioners at all levels in order to facilitate coordinated and timely deployment of multi-sectoral/jurisdictional resources when and where they are most needed.

The paper is quite interesting and merits publication. In the final version I would be happy if the authors will be able to answer to th following two questions.

Recalling the KTA process, the authors make reference to the concepts Ikujiro Nonaka (and Hirotaka Takeuchi) introduced in the studies about knowledge creation within organizations (Nonaka & Takeuchi, The knowledge creating company), but not their theories: this is strongly confusing. Why not make reference to their knowledge creating spiral?

In their paper, the authors never take into account the growing role the research of the responsible(s) has after disasters: I don't know if this is due to the fact that this is not true in Canada (in my country search of the responsible is becoming so invasive that it is a growing obstacle to learning form previous experiences): but discussing the fact that deep analysis of a disaster analyzing its causes and the errors in its management can be considered as an recognition of a bad behaviour could be important for non canadian readers.

Author Response

Recalling the KTA process, the authors make reference to the concepts Ikujiro Nonaka (and Hirotaka Takeuchi) introduced in the studies about knowledge creation within organizations (Nonaka & Takeuchi, The knowledge creating company), but not their theories: this is strongly confusing. Why not make reference to their knowledge creating spiral?

Thanks for the comment. To clarify these concepts, we added the following sentences: “Such concepts build notably on the “knowledge creation spiral” theory within organizations, introduced by Nonaka & Takeuchi in 1995, which emphasizes on the importance of involving both the top and front-line employees in knowledge creation process [16]. By integrating top-down and bottom-up approaches, this system enables the creation, accumulation and translation of tacit and explicit knowledge.” (Introduction, p.3) and added a new reference (Nonaka & Takeushi, 1995).

In their paper, the authors never take into account the growing role the research of the responsible(s) has after disasters: I don't know if this is due to the fact that this is not true in Canada (in my country search of the responsible is becoming so invasive that it is a growing obstacle to learning from previous experiences): but discussing the fact that deep analysis of a disaster analyzing its causes and the errors in its management can be considered as an recognition of a bad behaviour could be important for non Canadian readers.

The following sentence has been added to discuss this important obstacle to disaster knowledge management: “The search of the responsible(s) and/or the causes after disasters is negatively perceived in many countries, which may become a growing obstacle to learning from previous experiences.” (Results, p.9).

Reviewer 2 Report

Please see attached for some suggested edits and comments. Main issue is with the value of Table 2 which I find to be distracting and confusing from the main points and following discussions.  You may wish to either delete the table and just use the following discussion or consider just displaying the matrix graphically to help the reader understand the set-up for the following discussion. Overall the paper is nicely written, original, and raises important findings and ideas to help further understanding and ongoing efforts to improve environmental public health responses to disasters. 

Author Response

Please see attached for some suggested edits and comments.

All the suggested modifications have been added in the revised version of the manuscript.

Main issue is with the value of Table 2 which I find to be distracting and confusing from the main points and following discussions.  You may wish to either delete the table and just use the following discussion or consider just displaying the matrix graphically to help the reader understand the set-up for the following discussion.

Table 2 has been removed from the revised manuscript.
